# Spatio-Temporal Analysis of Marine Water Quality Data Based on Cross-Recurrence Plot (CRP) and Cross-Recurrence Quantitative Analysis (CRQA)

**DOI:** 10.3390/e25040689

**Published:** 2023-04-19

**Authors:** Zhigang Li, Ting Sun, Yu Wang, Yujie Liu, Xiaochuan Sun

**Affiliations:** 1College of Artificial Intelligence, North China University of Science and Technology, Bohai Road, Tangshan 063210, China; 2Hebei Key Laboratory of Industrial Perception, Tangshan 063210, China; 3Marine Ecological Restoration and Smart Ocean Engineering Research Center of Hebei Province, Qinhuangdao 066000, China

**Keywords:** marine water quality, CRP, CRQA, dynamic correlation, spatio-temporal analysis

## Abstract

In recent years, with the frequency of marine disasters, water quality has become an important environmental problem for researchers, and much effort has been put into the prediction of marine water quality. The temporal and spatial correlation of marine water quality parameters directly determines whether the marine time-series data prediction task can be completed efficiently. However, existing research has only focused on the correlation analysis of marine data in a certain area and has ignored the temporal and spatial characteristics of marine data in complex and changeable marine environments. Therefore, we constructed a spatio-temporal dynamic analysis model of marine water quality based on a cross-recurrence plot (CRP) and cross-recurrence quantitative analysis (CRQA). The time-series data of marine water quality were first mapped to high-dimensional space through phase space reconstruction, and then the dynamic relationship among various factors affecting water quality was visually displayed through CRP. Finally, their correlation was quantitatively explained by CRQA. The experimental results showed that our scheme demonstrated well the dynamic correlation of various factors affecting water quality in different locations, providing important data support for the spatio-temporal prediction of marine water quality.

## 1. Introduction

With the acceleration of industrialization, the discharge load of industrial, agricultural, and domestic sewage in coastal areas has increased, and the deterioration of marine water quality has worsened year by year. Red tide disasters and the eutrophication of water bodies occur frequently. The quality of marine water is not only important for the economy, but it also affects human life. Therefore, the analysis of marine water quality has become a hot topic for researchers in recent years. Industrialization, mining, pollution, and natural disasters affect water quality. They introduce or change various parameters of water, thus affecting whether water is suitable for human consumption or general use [1].

Because of the important role of the marine environment in social development, marine data analysis has always been a hot topic. Water quality is affected by many factors, such as meteorology, chemistry, and human activities, which leads to the nonlinearity, randomness, and delay of water quality parameters. Statistical learning and other methods are widely used in the analysis and modeling of marine data. Jiang et al. [2] used the parallel FP-growth algorithm to analyze the oxygen, temperature, phosphates, nitrates, and silicates in the ocean. Meanwhile, based on association rules, they analyzed the correlation between different data. Erni-Cassola et al. [3] used the meta-analysis method to detect increased marine plastic debris in the sea surface and explored the separation degree of plastic debris in the water according to the density of microplastic polymers, so as to further determine whether the plastic debris sank under the sea. Lemenkova [4] used a statistical database embedded in Python for data analysis to study the interaction between environmental factors affecting the seafloor geomorphology of the Mariana Trench, established a geospatial data model, and studied the inhomogeneity of the seafloor structure. Deng et al. [5] put forward a water quality analysis framework based on the time-series data-mining method, determined the relationship between water quality in the main stream and tributaries of the Yangtze River and the variation rule of dissolved oxygen, and effectively mined valuable knowledge from historical time-series data of water quality. However, the above research methods did not pay attention to the changes in the data across time and space, ignoring the effects brought about by cross-regional factors. Additionally, they lacked an analysis of the internal dynamic characteristics and spatial characteristics of the data. It is necessary to accurately analyze the non-stationarity and spatial heterogeneity of marine data.

A CRP can mine the correlation characteristics of time-series data in space-time, process nonlinear and non-stationary high-dimensional data, and realize the visualization analysis of marine data information. A CRP detects the correlation between signals by mapping two signals to the same phase space. The details of dynamic changes between two signals can be intuitively explained by the classical structure of points, diagonals, and rectangles. In addition, the traditional analysis method of a CRP is CRQA, which describes the CRP of signals in different states through nonlinear characteristic quantities such as the recurrence rate, certainty rate, recurrence entropy, and stratification rate. Both have excellent applications in disease diagnosis, industrial fault identification, financial market analysis, intelligent oceans, and many other fields. Kanakambaran et al. [6] used a CRP to analyze signals captured by fiber sensors in order to improve the accuracy of partial discharge detection and localization. González et al. [7] used a CRP and CRQA to analyze systolic blood pressure and inter-beat intervals in healthy persons and nephrotic patients. Wang et al. [8] applied a CRP and CRQA to the time recording of current and voltage fluctuations in electrochemical noise analysis to identify dynamic characteristics and achieve efficient detection results. Li et al. [9] demonstrated the dependent behavior of chlorophyll a with various marine factors using a CRP and CRQA and quantified their dynamic similarity using recurrence analysis. Currently, the use of CRPs and CRQA in the marine field mainly focuses on a single area and lacks the exploration of the spatio-temporal dynamic characteristics of marine data. In view of the advantages of CRPs and CRQA in many fields and in the analysis of marine environmental data, in this study, we used them to analyze the dynamic correlation of marine water quality in time and space.

In order to solve the problem that marine water quality data analysis mainly focuses on a single region and ignores the spatio-temporal impact across regions, this paper proposes a spatio-temporal dynamic analysis model of marine water quality data based on CRPs and CRQA. Due to the complex and changeable characteristics of the marine environment, various data not only interact with each other, but also interact with factors in different locations. Considering the influence of time and space, the accurate analysis of marine data is the key premise for water quality prediction. Therefore, we selected eight major factors from three stations to analyze the dynamic correlation among different stations. Specifically, the scheme consisted of phase space reconstruction (PSR), spatio-temporal dynamic CRP analysis, and CRQA quantification. Firstly, a marine factor of one station and influencing factors of the other two stations were mapped to the same phase space, and then their dynamic correlation was displayed through a CRP. Some key indicators of CRQA are used to evaluate the correlation of multiple marine water quality parameters. In particular, mutual information entropy can be used to measure the dependence between two discrete variables, so mutual information entropy is commonly used as an index to evaluate the correlation degree of marine factors. Finally, we could screen out the key factors affecting water quality. The results indicated that the scheme proved the interaction between marine data of different stations and effectively selected important factors, providing data support for the input of future marine prediction models. The key contributions of this work can be summarized as follows.

This was the first attempt to use CRPs and CRQA to study the dynamic association between water quality factors at different stations, as well as to analyze the factors in time and space.The water quality time-series data were converted to high-dimensional phase space through phase space reconstruction. After the reconstruction parameters were determined, the dynamic correlation relationship between the water quality data of different stations was displayed through a CRP, and the correlation degree was quantitatively described by CRQA and mutual information entropy.The spatial correlation of different influencing factors of water quality was confirmed, providing more reliable data support for the input of marine prediction tasks.

Specifically, the framework of the model is shown in Figure 1.

The rest of this paper is structured as follows. Section 2 summarizes some relevant concepts in the correlation analysis of water quality factors. In Section 3, the experimental results from the CRP and CRQA investigation of the dynamic correlation between water quality factors at different stations are presented, and the correlation between the main water quality factors is discussed and explained. Finally, Section 4 provides conclusions and prospective research directions.

## 2. Materials and Methods

Herein, a new spatio-temporal recurrence analysis model for the influencing factors of marine water quality is introduced in detail. Firstly, the model is summarized, and then the main components of the model are introduced, including PSR, CRP, and CRQA.

### 2.1. Overview

In order to find the factors affecting the dynamic correlation of marine water quality in time and space and determine the main influencing factors of the quality of the water, we set up an analysis model based on CRPs and CRQA. According to the investigation and analysis, we selected eight kinds of water-quality-influencing factors from three stations to analyze the spatio-temporal correlation between them. Specifically, our model consisted of the following three parts: PSR, CRP analysis, and CRQA quantification.

Taking pH as an example, firstly, the pH of the center station and the water-quality-influencing factors of the other three stations were mapped to the same phase space to calculate the reconstruction parameters. Secondly, the cross-recurrence matrix of the time series of the pH and another water quality factor was obtained through a CRP. Finally, some CRQA indicators were used to quantify the influence degree of each water quality factor on the pH. Based on the above experimental results, the temporal and spatial correlation between water quality factors was analyzed.

### 2.2. PSR Conversion

For chaotic time-series analysis, it is necessary to reconstruct the phase space formed by these sequence changes to explore the temporal dynamics of the system [10]. PSR can transform a one-dimensional time series to high-dimensional phase space. The reconstructed marine water quality time series could exhibit more nonlinear dynamic characteristics while preserving the continuity of the original series. PSR was considered as a prerequisite for CRPs in the task of the spatio-temporal correlation analysis of marine water quality factor data. Its purpose was to project the corresponding marine series into the high-dimensional phase space to obtain chaotic attractors. For the one-dimensional water quality time series w(t)= $w_{1},w_{2},\ldots ,w_{i}$ with a length of *i*, the reconstructed n-dimensional phase space could be expressed as follows:(1)$$w=\left[\begin{array}{c} w_{1}\\ w_{2}\\ \vdots \\ w_{N} \end{array}\right]=\left[\begin{array}{cccc} w_{1+t} & w_{1+2t} & \cdots & w_{1+(n-1)t}\\ w_{2+t} & w_{2+2t} & \cdots & w_{2+(n-1)t}\\ \vdots & \vdots & \vdots & \vdots \\ w_{N+t} & w_{N+2t} & \cdots & w_{N+(n-1)t} \end{array}\right]$$
where *N* = *i* + (n−1)*t*, *t* represents the delay time, and *n* represents the embedding dimension. In our paper, the mutual information method was considered to select the appropriate delay time, and the false nearest neighbors was used to select the suitable embedding dimension [11,12]. Note that unified reconstruction parameters should be selected so that two marine series can be reconstructed into the same phase space. A higher embedding dimension *n* and smaller delay time *t* were chosen in our scheme.

### 2.3. CRP Visualization

PSR is the first step in analyzing time series using a CRP [13]. After PSR, a CRP was used to detect the dynamic information of two marine water quality time series in the same phase space, and the data were visualized by two-dimensional graphics. The CRP was determined by a cross-recurrence matrix. For two reconstruction variables $\vec{a}$ and $\vec{b}$, the matrix is defined as follows: (2)$$CR_{i,j}^{{\vec{a}}_{i}\cdot {\vec{b}}_{j}}\left(\varepsilon \right)=\Theta \left(\varepsilon -∥{\vec{a}}_{i}-{\vec{b}}_{j}∥\right),i=1\ldots N,j=1\ldots M$$
(3)$$\Theta \left(e\right)=\left\{\begin{array}{c} 1,e\geq 0\\ 0,e<0 \end{array}\right.$$
where *N* and *M* denote the length of ${\vec{a}}_{i}$ and ${\vec{b}}_{j}$, respectively; ε is the threshold that is 0.6% of the maximum phase space diameter [11]; and $\Theta \left(•\right)$ represents the Heaviside function.

$CR_{i,j}$ is a two-dimensional matrix containing 1 and 0. When the distance between two sequences in the same phase space is less than ε, $CR_{i,j}$ is 1 and represented as a black point in the CRP; otherwise, $CR_{i,j}$ is 0 and represented as a white point in the CRP. Matching black and white dots with 1 and 0 were used to visualize similar behavior between two time series.

### 2.4. CRQA Quantification

In the analysis of marine time-series data, we used CRQA to quantify the frequency of similar changes between the time series of two water quality factors and then confirmed the dynamic spatial correlation of the time series of different water quality factors. CRQA indexes include mean diagonal line length (MDL), determinism (DET), laminarity (LAM), and recurrence rate (RR) [14,15,16,17,18].

RR, a metric for the density of recurrence points, exposes the probability that the time series of water quality factors remain similar under a certain delay. A high RR value indicates a high probability of similar states between both time series. It is determined by
(4)$$RR\left(\varepsilon \right)=\frac{1}{N^{2}}\sum_{i,j=1}^{N}CR_{i,j}\left(\varepsilon \right)$$

DET represents the ratio of diagonal structures. Usually, a high DET implies a deterministic process, while a low DET suggests a random process. In our case, it could evaluate the regularity and predictability of the interaction between two marine time series, expressed by
(5)$$DET=\frac{\sum_{l=l_{\min}}^{N}lP\left(l\right)}{\sum_{l=1}^{N}lP\left(l\right)}$$
where $l_{\min}$ is set to 2 [19], and P(l) denotes the histogram of diagonals, defined as
(6)$$\begin{array}{cc} P(l)= & \sum_{i,j=1}^{N}\left(1-CR_{i-1,j-1}\right)\left(1-CR_{i+l,j+l}\right)\\ \; & \times \prod_{k=0}^{l-1}CR_{i+k,j+k} \end{array}$$

LAM is the percentage of recurrence points comprising vertical structures. It hints at the steady states between two marine water quality time series, calculated as follows:(7)$$LAM=\frac{\sum_{v=v_{\min}}^{N}vP\left(v\right)}{\sum_{v=1}^{N}vP\left(v\right)}$$
where $v_{\min}$ is set to 2, and P(v) denotes the histogram of verticals, given by
(8)$$\begin{array}{cc} P(v)= & \sum_{i,j=1}^{N}\left(1-CR_{i-1,j-1}\right)\left(1-CR_{i,j+v}\right)\\ \; & \times \prod_{k=0}^{v-1}CR_{i,j+k} \end{array}$$

MDL refers to the mean length of diagonal lines, and its value is the average time for which the time series of the marine water quality factors in the two groups are similar, given by
(9)$$MDL=\frac{\sum_{l=l_{\min}}^{N}lP\left(l\right)}{\sum_{l=l_{\min}}^{N}P\left(l\right)}$$

### 2.5. Mutual Information Entropy

Information entropy is a measure of the uncertainty of random variable X in statistics. The higher the uncertainty of X, the greater the entropy. For a discrete random variable X, its probability distribution is consistent with p(x)=P(X=x), where x∈X. Information entropy is defined as follows:(10)$$H\left(X\right)=-\sum_{x\in X}p(x)\log p(x)$$

For two discrete random variables X and Y, the joint distribution probability is p(X,Y), and the marginal distribution probability is p(X), p(Y), where x∈X and y∈Y. Then, according to the definition of information entropy, the joint distribution entropy is
(11)$$H\left(X,Y\right)=-\sum_{x\in X}\sum_{y\in Y}p(x,y)\log p(x,y)$$

Mutual information entropy is used to measure the degree to which one random variable X reduces the uncertainty of another random variable Y. The mutual information entropy between X and Y can be defined as
(12)I(X;Y)=H(X)+H(Y)−H(X,Y)

## 3. Results

In this section, we present a thorough experimental evaluation of marine water quality within the framework of CRPs and CRQA. Firstly, we introduce the sources of the marine water quality data used in this study, followed by the determination of the PSR parameters. Secondly, a CRP was used to visualize the spatio-temporal characteristics of the marine water quality parameters at the three stations. Finally, some key CRQA indicators and mutual information entropy were used to quantify the degree of the temporal and spatial correlation of these parameters. We also explain the importance of the above research work for future marine water quality prediction tasks.

### 3.1. Datasets

The data used in this experiment came from a region in the Bohai Sea of China, and the data were collected by land-based stations and ocean buoys. A total of 4320 samples were collected from 22 July to 20 October 2021, with an interval of 30 min. After consulting the data, eight factors affecting the marine water quality were selected, including dissolved oxygen (Do), chlorophyll a (Chl), turbidity (Turb), blue-green algae (Bga), total dissolved solids (Tds), dissolved oxygen saturation (DoP), water temperature (Temp), and pH [20,21,22,23,24,25]. In order to explore the spatio-temporal characteristics of the marine water quality factors, the water quality factors were gathered from three adjacent stations. The locations and distances are shown in Figure 2. Specially, due to the large number of parameters calculated, the following example uses pH as the key indicator of marine water quality in order to illustrate the experimental process.

### 3.2. Parameter Determination

Before CRP was used for the dynamic correlation analysis of the marine water water quality data, one-dimensional time-series data of water quality had to be mapped to high-dimensional space, which was achieved through PSR. The delay time and embedding dimension had to be calculated. Taking pH as an example, the dynamic correlation between other water quality factors and the pH value was analyzed. Table 1 and Table 2, respectively, show the embedding dimension and delay time of each water quality factor in the three stations. Table 3 shows the optimal matching of PSR parameters when pH and other water quality time series were reconstructed into the same high-dimensional space. A larger embedding dimension and a smaller delay time were generally considered the best matches.

### 3.3. Spatio-Temporal Visualization and Results

A CRP contains many classical structures to analyze the correlation of series, including recurrence points, diagonal lines, and vertical/horizontal diagonal distribution. Figure 3, Figure 4 and Figure 5 show the correlation between the pH at ST1 and the other water quality factors at ST1, ST2, and ST3. Taking pH as an example, we qualitatively determined the correlation between the water quality time-series data of the three stations and the pH of the central station, as shown in the figures below.

Figure 3 shows the correlation analysis between the pH and the other factors of the central station (ST1). It can be seen from the black recurrence points, rectangular structures, vertical horizontal lines, and other CRP structures in Figure 3 that all factors had a certain degree of correlation with the pH. Specifically, there are significantly more CRP structures in Figure 3d and Figure 3e,g than in Figure 3b,c. This indicates that the similarity of the recurrence behavior between Bga, Tds, and Temp and pH was stronger than that for Chl and Turb in ST1. The latter had a weak correlation with pH.

Figure 4 and Figure 5 show the analysis results of the temporal and spatial correlation between the water quality factors at ST2 and ST3 and the pH at the central station (ST1), respectively. It can be clearly seen that the factors in Figure 4 and Figure 5 also had similar characteristics to those in Figure 3. Specifically, there are a large number of rectangular structures in Figure 4e and Figure 5e, and there are many horizontal line segments in Figure 4g and Figure 5g. Nevertheless, in Figure 4b and Figure 5b, there are few black recurrence points and large areas of blank space. This reflects a more synchronized state between the Tds and Temp at ST2 and ST3 and the pH at the central station. The pH and Chl at ST1 did not present obvious behavioral similarities. The above visualization results were consistent with those shown in Figure 3. By analyzing the recurrence behavior of factors among multiple stations, the temporal and spatial dynamic correlation characteristics of the marine water quality factors were verified.

### 3.4. CRQA Analysis

As a quantitative analysis method of CRPs, CRQA includes several indicators to measure the influence degree of factors including RR, DET, MDL, LAM. In this study, we analyzed the influence degree of water quality factors through these four indicators and quantitatively evaluated the correlation of water quality factors by integrating the four indicators. Table 4 shows the quantification results of the four CRQA indexes for the correlation between seven water quality factors at three stations and pH at the central station.

RR indicates the probability that two water quality sequences had similar states. In the central station ST1, the RR value of Tds was 0.0727, which was significantly higher than that of the other water quality factors, indicating more similar recurrence states between the Tds and pH series. The RR value of Chl was only 0.0114, indicating a very weak correlation between the CHL and pH sequences. We also reached the same conclusion for the ST2 and ST3 stations, which indicated that the Tds at the ST2, ST3, and ST1 stations had a highly similar impact on the pH of the ST1 station. Similarly, Chl had the lowest RR values among the three stations, indicating that there was no obvious correlation between Chl and pH when analyzing the impact of a single station or multiple stations.

As an important measure of the similarity of two time series, MDL reveals the average duration of similar states in the phase space of two time series. The values of Tds were 6.5991, 8.8260, and 7.0533, and the values of Temp were 9.0706, 7.3402, and 8.5231. These were still higher than the values for other water quality factors. The Chl MDL values were 3.4141, 3.6056, and 3.5846, representing the worst performance.

DET characterizes the determinacy of two sequences by calculating the ratio of the diagonal lines in the CRP. When two deterministic processes have similar phase space states, the corresponding DET value is larger. At the three stations, the values of Tds were 0.9355, 0.9513, and 0.9387, and the values of Temp were 0.9562, 0.9582, and 0.9531, respectively. The DET values of the two were significantly higher than those of the other factors, indicating that their states were more similar to the phase space of the pH sequence, while the findings for Chl were the opposite.

LAM represents the synchrony between sequences by quantifying the vertical/horizontal structures. From Table 4, it can be seen that the Tds and Temp values were the largest, while the Chl value was the smallest. This implied that Tds and Temp were more in sync with the pH sequences, and Chl was almost out of sync with pH.

### 3.5. Quantitative Analysis

In Section 3.4, we found that under the four indexes, the water quality factors did not have exactly the same degree of influence on pH. For example, at ST2, the value of Turb was higher than that of Bga considering the influence of MDL, but the opposite was true at ST3. For LAM, the Do and DoP values also presented the same performance across stations. A single index cannot measure the effect of several water quality factors on another factor. Therefore, we added the value of mutual information entropy, normalized it with the four CRQA indexes, calculated the geometric mean value, and comprehensively measured the behavioral similarity among the water quality factors. By introducing mutual information entropy, the correlation between the data could be mined from the perspective of recurrence analysis and information theory, which made the scheme more convincing.

Figure 6 shows the comprehensive evaluation results of CRQA and mutual information entropy for the spatio-temporal correlation characteristics of the marine factors. The figure in the heat map is the combined value of the correlation between the two factors, ranging from 0 to 1. The larger the value, the stronger the temporal and spatial correlation between the two factors. At the same time, the colors reflect the degree of correlation along with the values. The darker the color, the stronger the relationship between the two factors. Taking Figure 6h as an example, considering their impact on the pH of the central station, the top four factors at ST1 were Temp, Tds, DoP, and Turb. At ST2, they were Temp, Tds, Do, DoP. At ST3, they were Temp, Tds, DoP, and Turb. In the case of interference by various factors, we could assume that the high-correlation factors of the three stations were basically the same. In the pH prediction task, we could select the water quality factors of the target station and the high-association factor data of the adjacent stations as the input. High-correlation factors for other water quality factors can also be found in Figure 6. In marine water quality multi-task prediction, one can conduct a comprehensive evaluation of water quality by predicting various factors affecting water quality at the same time, so that the prediction results are more convincing. In addition, the CRQA comprehensive measurement values corresponding to the marine water quality factors in Figure 6 are presented in Table A1.

Furthermore, in order to verify the scheme for multiple data lengths, we chose 2000 and 3000 pieces of data to draw a comparison with the analysis results of the total 4320 pieces of data. Figure 7 shows the correlation between the pH sequences for the three data lengths and other factors. As can be seen from the values and color shades in this figure, in the experiments with data lengths of 2000 and 3000, the correlation between the pH sequence and Tds and Temp was still large, while that for Chl was small. This was almost consistent with the results for 4320 pieces of data.

## 4. Conclusions

In this paper, we developed a spatio-temporal analysis model of marine water quality data based on CRPs and CRQA. Furthermore, mutual information entropy was introduced as one of the evaluation indexes. In contrast to single-station analysis, this model could analyze the spatio-temporal dynamic characteristics of water quality factors at multiple stations. Accordingly, we could obtain better prior data as the input for subsequent marine water quality forecasting tasks. Through spatio-temporal analysis, it could be concluded that certain variation factors of the target station were largely affected by other factors at adjacent stations. In future marine water quality multi-tasking prediction work, we will give full consideration to the effects of other stations and consider comprehensive data changes caused by many factors, so as to improve the accuracy of prediction and provide a more novel research scheme for marine prediction work.

## Figures and Tables

**Figure 1 entropy-25-00689-f001:**
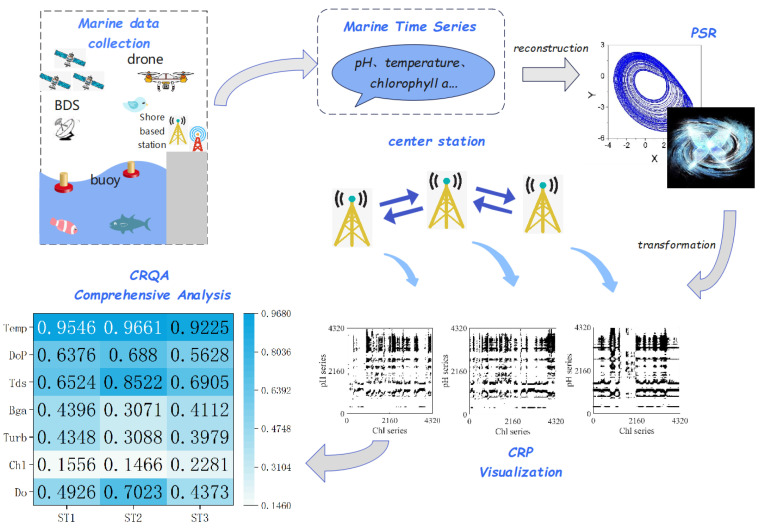
Marine water quality spatio−temporal analysis framework based on CRPs and CRQA.

**Figure 2 entropy-25-00689-f002:**
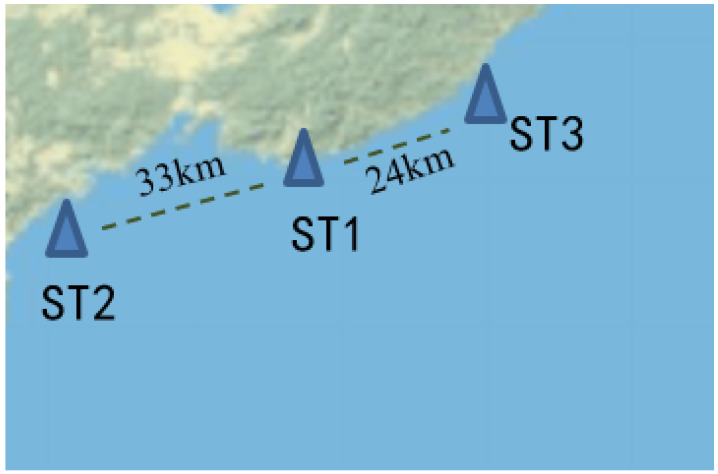
Geographical location of the three marine monitoring stations.

**Figure 3 entropy-25-00689-f003:**
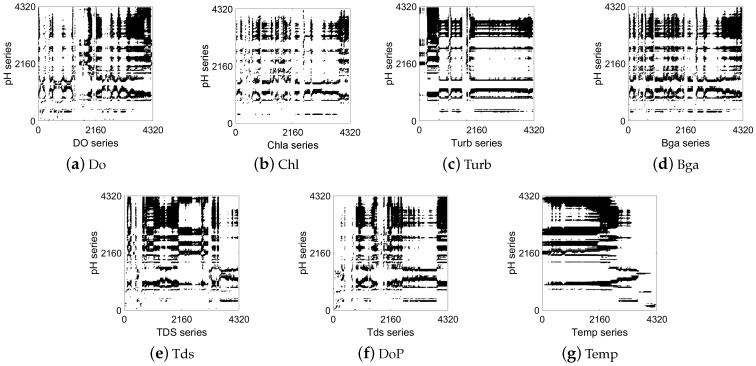
CRP visualization of correlation between marine factors and pH at ST1 station.

**Figure 4 entropy-25-00689-f004:**
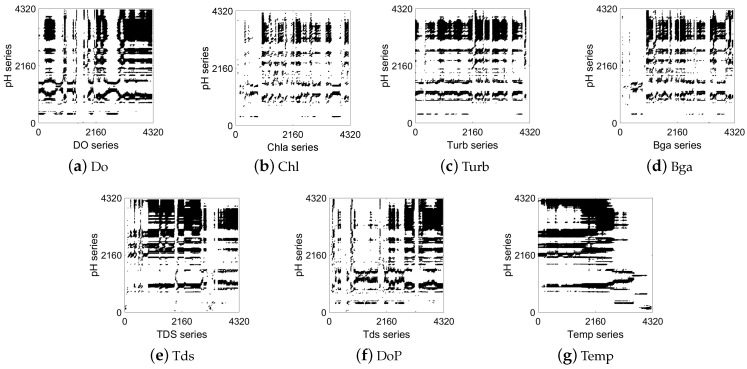
CRP visualization of correlation between marine factors at ST2 station and pH at ST1 station.

**Figure 5 entropy-25-00689-f005:**
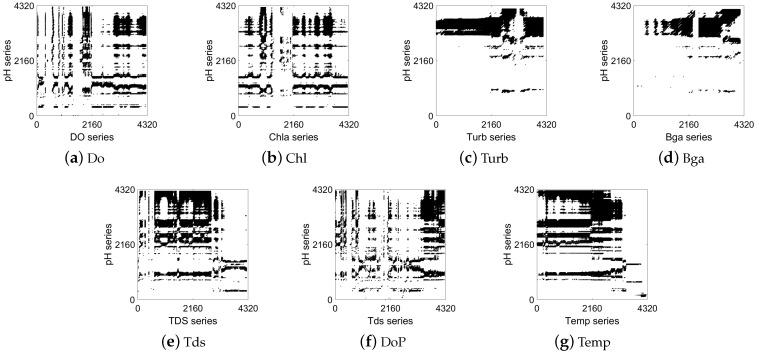
CRP visualization of correlation between marine factors at ST3 station and pH at ST1 station.

**Figure 6 entropy-25-00689-f006:**
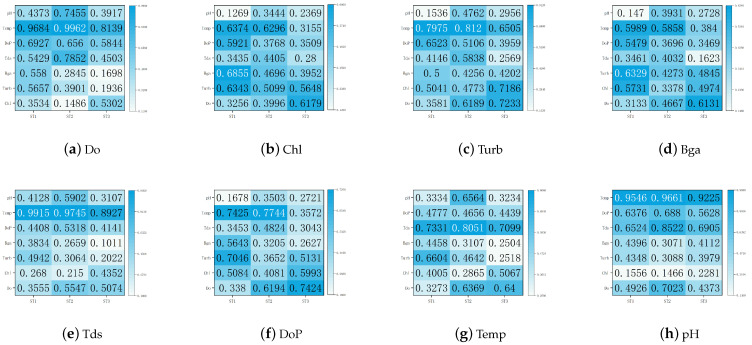
CRQA indicators and mutual information entropy comprehensive measurement results of the associations among marine factors.

**Figure 7 entropy-25-00689-f007:**
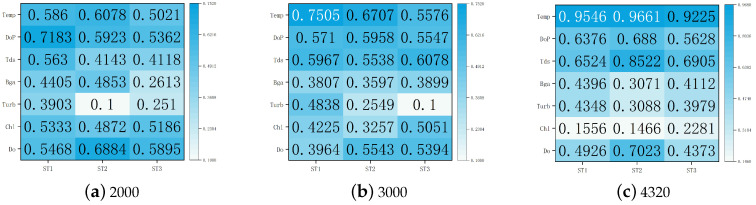
Comprehensive measurement results of correlation between pH sequences and other factors for different data lengths.

**Table 1 entropy-25-00689-t001:** Determination of embedding dimension of water quality sequences at the three stations.

Station	Do	Chl	Turb	Bga	Tds	DoP	Temp	pH
ST1	5	8	6	5	6	5	5	6
ST2	4	9	8	8	7	6	5	6
ST3	4	7	15	17	7	5	6	5

**Table 2 entropy-25-00689-t002:** Determination of delay time of water quality sequences at the three stations.

Station	Do	Chl	Turb	Bga	Tds	DoP	Temp	pH
ST1	18	28	42	22	19	14	14	9
ST2	22	18	22	22	35	15	15	18
ST3	13	15	24	24	20	21	26	18

**Table 3 entropy-25-00689-t003:** Taking pH as an example, the best-match results of embedding dimension/delay time.

Station	Do	Chl	Turb	Bga	Tds	DoP	Temp	pH
ST1	6/9	8/9	6/9	6/9	6/9	6/9	6/9	6/9
ST2	6/9	9/9	8/9	8/9	7/9	6/9	6/9	6/9
ST3	6/9	7/9	15/9	17/9	7/9	6/9	6/9	6/9

**Table 4 entropy-25-00689-t004:** CRQA quantification results of associations between seven factors from three stations and pH at the central station.

Station	CRQA	Do	Chl	Turb	Bga	Tds	DoP	Temp	pH
ST1	MDL	5.2637	3.4141	5.2753	4.7075	6.5991	5.6745	9.0706	5.5753
	RR	0.0411	0.0114	0.0570	0.0386	0.0727	0.0452	0.0452	0.0602
	DET	0.8994	0.7758	0.8859	0.8684	0.9355	0.9063	0.9562	0.8940
	LAM	0.9281	0.8402	0.9087	0.9097	0.9446	0.9234	0.9574	0.9384
ST2	MDL	7.6584	3.6056	4.1526	3.9768	8.8260	0.7376	7.3402	9.1455
	RR	0.0848	0.0093	0.0339	0.0194	0.1029	0.0590	0.1082	0.1635
	DET	0.9398	0.7600	0.8166	0.8312	0.9513	0.9333	0.9582	0.9745
	LAM	0.9443	0.8580	0.8896	0.8874	0.9538	0.9417	0.9589	0.9866
ST3	MDL	4.7679	3.5846	5.8109	6.1439	7.0533	6.6590	8.5231	5.2118
	RR	0.0330	0.0218	0.0566	0.0317	0.0698	0.0401	0.1120	0.0456
	DET	0.8826	0.8180	0.8617	0.9069	0.9387	0.9222	0.9531	0.9236
	LAM	0.8992	0.8701	0.9225	0.9310	0.9501	0.9331	0.9553	0.9591

## Data Availability

The datasets for this study are available upon request from the corresponding author.

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
