# Peer review of "Spatio-Temporal Analysis of Marine Water Quality Data Based on Cross-Recurrence Plot (CRP) and Cross-Recurrence Quantitative Analysis (CRQA)"

_entropy, 2023, doi:10.3390/e25040689_

Round 1

Reviewer 1 Report

The paper addresses a statistical analysis on marine water quality data. Several methods are presented and applied to eight water parameters registered at three oceanographic stations. The main drawback of the paper is to present the methods and their outputs but without inputs description. It renders difficult to assess the qualities of the methods and of the outputs. Consequently, we suggest asking for a large revision of the paper. Here follows some suggestions from different points of view.

Science

Eq(1) is valid only if t=0.

Eq(9) for MDL is the eq(5) for DET.

The absence of salinity in the parameters could be detailed.

The distance between the three stations could be given.

An analysis of the precision of the methods could be made by comparing with less than whole 4320 records.

The dynamic property may be mandatory in the example, because the end of the summer 2021 is considered as a whole in the 4320 records used. Spatial dimension is convincing in fig (5).

There is no details on pollution sources, on stations and sampling methods, and water domain kinds (marsh, estuary…).

Edition

The dynamic aspect remains to be developed in the results.

“weather and human”, at line 40, may be edited.

The bibliographic examples given in the introduction address mainly other topics like imagery, industry and health. The references could review the marine environment.

“water quality” appears 18 times p 3, which may be excessive.

The location of the stations could be detailed. The 3 stations could be described (buoy, drone…).

The difference between “recursion entropy” line 52 and “information entropy” line 98 could be clarified.

Check, may be 2*H(X,Y) in eq(12).

Figures are clear. Units in Tables must be detailed.

Typos

Missing space, lines 22, 26, 60, 254

Missing dot, line 62

Missing reference, line 68

Singular at “station” line 197

Table 5 is 4, line 234

Small characters: We, line 236, Marine, line 255

Plural may be at “stations”, line 250

Edit ref 17.

Capital characters at Bengal and Bangladesh in ref 24.

Indicate city in refs 9, 14, 29.

Reviewer 2 Report

The manuscript aims to show the usefulness of Cross recursive Analysis applied to the study of marine environmental water quality in spatial and time series.

First, since the abbreviations CRP and CRQA are not widely known, they should not be abbreviated in the title of the manuscript and should be written with the full name.

In the introduction, the sense that the authors want to express is understood; but the wording of the first paragraph is not correct, especially when talking about the use or human consumption of seawater. It is suggested that they better specify which water uses are spoiled by pollution (e.g., aquaculture or recreational use).

In the second paragraph (lines 25-46) they should also consider the alteration of quality due to the mouth of rivers at certain coasts as an element that modifies the characteristics, without human influence being considered in many cases, especially in large rivers.

In general, the introductory part cites several bibliographical references on studies related to the subject; but the way of writing is a bit detached, limited to a list of names and brief description of the content of the work, but without showing a relationship between them. It would be convenient to write showing the link between the references presented (lines 47-84).

The methodology is correctly presented; but the metrics used in section 2.4 are not referenced to any academic work. It is as if they were developed by the authors, when they are standard procedures in the study of data series. It would be necessary to indicate several references in this section.

In the results, (line 227-233) similar behavior in the time series is commented; but it is not shown in any way with statistical test or figure that indicates what is expressed, it would be interesting to provide some additional information.

It would be interesting to discuss the results presented in Figure 5 with other academic works on the aquatic environment that show that these relationships presented here are similar to those that the usual works on water quality in the aquatic environment find with other procedures. For example, relationships between chlorophyll, temperature and turbidity.

Finally, the list of references is not presented in journal style and should be revised in a later version of the manuscript.

Reviewer 3 Report

This manuscript aims to study the dynamic association between water quality parameters of some marine time series data, based on Cross Recursive Plot (CRP) and Cross Recursive Quantitative Analysis (CRQA), and to analyze water quality parameters in time and space.

In the Introduction section, several citations do not follow the journal's guidelines and others do not appear in the List of References either.

Overall, this manuscript is a little careless and several parts should be carefully reworked, especially the Introduction and Results sections.

The analytical formulation (Materials and Methods section) is well-known and widely used. Many more interesting and useful references, including some with built-in MATLAB CRP and CRQA functions, should be provided, especially in subsections 2.3 and 2.4. References [22] and [23] are not enough.

Data series from three stations are mentioned several times throughout the manuscript and are used to show correlations between various water quality parameters at the stations. However, neither overall water quality characteristics nor the location of stations are given. A better characterization of the area, its uses, and (possible) sources of pollution should be provided.

Figures 2-4 are very poor and not explanatory enough; possibly expanding those figures and/or improving their content would help.

Table 5 is cited/mentioned in subsection 3.4 but is not provided. 

All figures together with their captions must be self-explanatory. Not all of them fulfill this request. Especially, the Figures 2-4 and 5 captions must describe accurately and fully clarify what the figures depict. Color legends would possibly help as well.

Finally, the English language is moderate. Check all parts of the manuscript and correct grammatical constructions. The authors should ask the help of a native English-speaking proofreader because there are some linguistic mistakes that should be fixed.

Round 2

Reviewer 1 Report

We suggest publishing the paper. The authors provide a very good revision in accordance with the review report items.

We mention only few details to be check:

Line 41: may be replace DO by Dissolved Oxygen.

In Figure 1: add CRQA in the figure.

Check in the formula for N: i-… (is it i+…?) line 134.

Remove “we use” line 156

Complete “and Results” in title section 3.3

Line 283 colorS  > colors

Reviewer 2 Report

The authors have modified the manuscript following the proposals of my review. 

However, I note that the paragraphs beginning on line 274 and 279 have the same beginning; perhaps this is an error and they could check the meaning of the two sentences in case it is necessary to modify it.

Reviewer 3 Report

Overall, some of my relevant concerns have been addressed satisfactorily. 

The text has been improved with the clarifications provided. Appropriate references and two new figures were considered. The complementary explanations about the content of Figures 6 and 7 are pertinent. Basically, the current scientific content of the manuscript is acceptable. However, there are still some inconsistencies that must be overcome.

As reported in the previous review, several citations do not follow the journal's guidelines.

When a publication is cited by its first author, whether or not including "et al.", the author's name must be followed by the reference number. Just a few examples: should be Jiang et al. [2] used the..., Erni-Cassola et al. [3] used the...,  Lemenkova [4] used the...Deng and Wang [5] put forward... (lines 29,32,35,38...), among many others throughout the manuscript. 

Several acronyms are not defined (written with the full name spelled out) in the core of the manuscript (the title and abstract are excluded); examples are CRP, CRQA, PSR, DET, LAM, RR, etc. All acronyms must be defined when they appear for the first time throughout the manuscript. Consider a list of acronyms/abbreviations; that would help the readers.

With some effort and following the text it is possible to detect some strong and weak correlations between the parameters, but Figures 3-5, including their captions, are not by themselves clearly explanatory. As reported in the previous review, these figures were expected to improve.

There are other issues that have not been fully answered, but I do not consider such gaps to be absolutely mandatory. However, careful spell-checking is strongly recommended.
